# Planum temporale asymmetry in newborn monkeys predicts the future development of gestural communication's handedness

Yannick Becker[1,4], Romane Phelipon[1], Damien Marie [1], Siham Bouziane[1], Rebecca Marchetti[1], Julien Sein[2], Lionel Velly[2], Luc Renaud[2], Alexia Cermolacce[3], Jean-Luc Anton[2], Bruno Nazarian[2], Olivier Coulon[2] & Adrien Meguerditchian [1,3] ✉

The planum temporale (*PT*), a key language area, is specialized in the left hemisphere in prelinguistic infants and considered as a marker of the pre-wired language-ready brain. However, studies have reported a similar structural *PT* left-asymmetry not only in various adult non-human primates, but also in newborn baboons. Its shared functional links with language are not fully understood. Here we demonstrate using previously obtained MRI data that early detection of *PT* left-asymmetry among 27 newborn baboons (*Papio anubis*, age range of 4 days to 2 months) predicts the future development of right-hand preference for communicative gestures but not for non-communicative actions. Specifically, only newborns with a larger left-than-right *PT* were more likely to develop a right-handed communication once juvenile, a contralateral brain-gesture link which is maintained in a group of 70 mature baboons. This finding suggests that early *PT* asymmetry may be a common inherited prewiring of the primate brain for the ontogeny of ancient lateralised properties shared between monkey gesture and human language.

Language is a prominent feature of the human species[1]. Nonhuman primates, however, share some "domain general" cognitive properties that are essential for language processes[1]. Whether these shared cognitive properties of humans and nonhuman primates are the result of continuous or convergent evolution can be examined by comparing their respective underlying structure: the brain. A key structure for the language network in the brain is the planum temporale (*PT*), a part of Wernicke's area[2]. This region has the particularity of presenting left-hemispheric specialisation, not only at the functional level in a variety of language processing tasks, but also at the structural level, with the size of the *PT* being larger in the left than in the right hemisphere in most adult humans[3]. Its specific adaptative value for language is unclear, although lateralisation may constitute a fitness advantage. When it comes to the brain, there is a significant optimisation in the accessible neural resources, especially for highly demanding cognitive resources like language. One hemisphere can perform a task while simultaneously, the other hemisphere oversees another task[4].

Interestingly, *PT* left asymmetry has also been found very early in infant development, both at the structural[5] and functional level[6]. It has been suggested that such early features of language brain lateralisation in infants may represent a human-specific prewired brain for language acquisition[6]. The latter hypothesis has been questioned by recent studies reporting similar structural *PT* asymmetries not only in different primate species[3,7,8] but also recently in newborn baboons[9,10], indicating that this brain feature is not language- or human-specific. It may also suggest a shared signature of a common ancient cognitive process at the heart of language evolution[7–10]. However, the function of such a structural *PT* asymmetry in monkeys remains unclear.

[1]LPC UMR7290 & CRPN UMR7077 Aix-Marseille Univ, CNRS, Marseille, France. [2]INT, UMR7289, Aix-Marseille Univ, CNRS, Marseille, France. [3]Station de Primatologie UAR 846, CNRS, CELPHEDIA, Rousset, France. [4]Present address: Department of Neuropsychology, Max Planck Institute for Human Cognitive and Brain Sciences, Leipzig, Germany. ✉e-mail: adrien.meguerditchian@univ-amu.fr

A potential relevant functional candidate to test is gestural communication given its shared "domain-general" intentional/goal-directed and referential properties with language[3,11,12]. Interestingly, just like in humans and chimpanzees[12–14], baboons' communicative gestures elicited specific manual lateralisation patterns in comparison to non-communicative manual activities (see Fig. 1). Namely, handedness for communicative gestures showed not only an increased use of the right hand at the population level but also different patterns of manual lateralisation at the individual level (for a review:[12]). Such specific manual lateralisation for gestural communication—but not handedness for non-communicative actions—was found associated with brain asymmetries of homologous key language structures, such as both Broca's area and *PT* in adult chimpanzees[7,15,16] and Broca's area in adult baboons[17].

In the present in vivo MRI and behavioural study in 97 baboons (*Papio anubis*) living in social groups, we tested whether structural *PT* asymmetries classification (leftward *versus* non-leftward) from our previous published MRI works[8,10], correlates with handedness measures for gestural communication in comparison with handedness for non-communicative actions. For Study 1, we did so longitudinally in 27 unweaned newborns from 4 days old, for which *PT* grey matter volume was already measured at birth[10] and for which hand preference measures were collected later in development (from 7–9 months old) as gestural and manipulative behaviours emerged. For Study 2, we did so in 70 mature baboons—including mostly adults but also weaned juveniles and adolescents—for which both *PT* surface[8] and behavioural measures were available at the time/age class of the MRI scanning.

Here we show that early detection of *PT* left-asymmetry in newborn baboons predicts the future development of right-hand preference for communicative gestures once juveniles but not for non-communicative actions. In addition, we found that this contralateral *PT*-gesture lateralisation link is maintained in the independent group of mature baboons. This finding suggests that early *PT* asymmetry may not be only considered as the human-specific marker for language development. It may rather be a common inherited prewiring of the primate brain for the ontogeny of ancient lateralised communicative properties, shared between baboon gesture and human language.

## Results

Following positive normality test, one-tailed two-sample *t*-tests showed a significant contralateral difference of group-level handedness (Mean Handedness Index, MHI) between left-biased *PT versus* non-left-biased *PT* groups in both newborn (Study 1: Fig. 2, $p < .036$) and mature baboons (Study 2: Fig. 3, $p < .015$) for gestural communication only but not for non-communicative manipulation ($p > .10$). Additional one-tailed one-sample *t*-tests highlighted a significant degree of positive MHI (i.e., right-handedness) in the left-biased *PT* group in both newborn (Study 1: Fig. 2, $p < .005$) and mature baboons (Study 2: Fig. 3, $p < .0015$) for gestural communication only, but not for

non-communicative manipulation or the non-left-biased *PT* groups ($p > .10$).

Further, for Study 1, a logistic regression was performed to investigate the effects of the *PT* lateralisation classification (i.e. left-biased *PT versus* non-left-biased *PT* groups) at birth on the likelihood of becoming left or right-handed in communicative gesture when juvenile, controlling for non-communicative handedness (Fig. 4A). The predictor variable "*PT* lateralisation classification" was found to contribute to the model. The unstandardised Beta weight for the predictor variable; $B = (-2.06)$, $SE = 1.2$, $Wald = 2.924$, $p < .044$. For non-communicative actions, the predictor variable "*PT* lateralisation classification" was not found to contribute to the model (Fig. 4B). Similarly, for Study 2, logistic regression was performed to investigate the effects of the "*PT* lateralisation classification" in mature subjects on the likelihood of being left or right-handed in communicative gesture, controlling for non-communicative handedness. The predictor variable "*PT* lateralisation classification" was found to contribute to the model. The unstandardised Beta weight for the predictor variable; $B = (-0.842)$, $SE = 0.3965$, $Wald = 4.515$, $p < .016$. For non-communicative actions in mature subjects, the predictor variable "*PT* lateralisation classification" was not found to contribute to the model.

## Discussion

The results of our studies in both 27 newborn and 70 mature baboons are straightforward:

In our cohort of 70 mature baboons (Study 2), we found that communicative gesture right-handedness, but not handedness for non-communicative manipulation, is associated with leftward structural *PT* asymmetry. This finding is consistent with previous research showing a correlation between handedness measures for gestural communication specifically and lateralised "language-homologue" brain structure, including the *PT* in chimpanzees[7,15] but also other regions of interest such as Broca's area in both chimpanzees[15,16] and baboons[17]. For instance, baboons preferring to communicate with their right hand have a deeper left-than-right inferior arcuate sulcus in its most ventral section (a sulcal border of Broca's area homologue) than those preferring to communicate with their left hand and vice versa[17]. However, those studies, including ours in the mature baboons, were conducted within a single time point/age class, namely when both manual behaviours and brain structures were already fully lateralised and mature. It was thus impossible to determine whether these brain asymmetries precede or follow the development of lateralised manual behaviours.

Our study in the cohort of 27 developing baboons at two longitudinal time points from birth (Study 1) allowed us to address this question by testing this "egg-chicken" hypothesis. We found that early detection of structural *PT* asymmetry at the first time point, namely soon after birth, predicts, for time point 2, the

Communicative Gesture          Non-communicative Actions

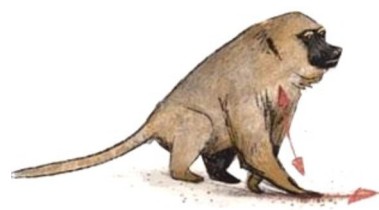
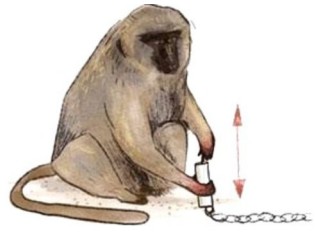

**Fig. 1 | Illustration of communicative gesture versus non-communicative actions in baboons.** The communicative gesture (on the left) consists of slapping or rubbing the hand on the ground to threat a conspecific or a human. The non-communicative actions (on the right) consists of a bimanual coordinated manipulation, one 'dominant' hand removes the food from a tube held by the other 'subordinated' hand.

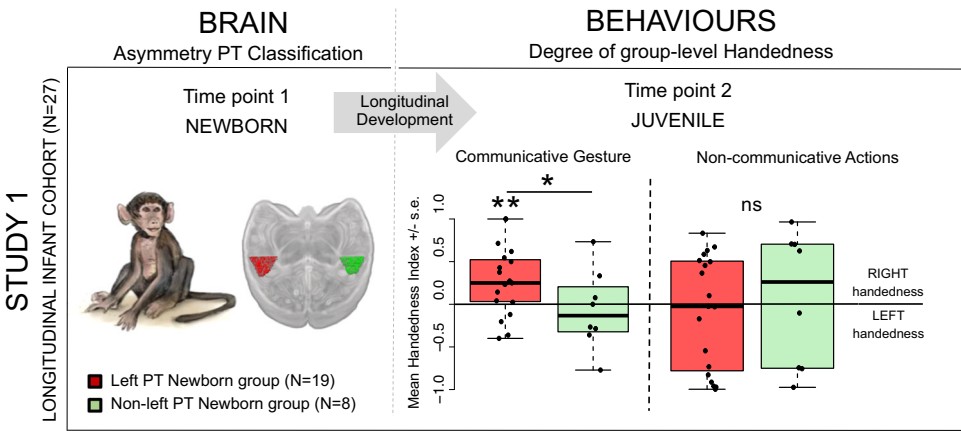

**Fig. 2 | Gestural communication handedness in juvenile baboons according to their early newborn planum temporale (*PT*) asymmetry.** Difference in the degree of group-level handedness (M.HI) in weaned juveniles (i.e., Time point 2) for the communicative gesture and for non-communicative actions, between left-biased *PT* (*N* = 19 in red) *versus* non left-biased *PT* (*N* = 8 in green) groups as early classified at birth (i.e. Time point 1). Positive values indicate right-hand preference, and negative values indicate left-hand preference. One-tailed two-sample *t*-tests showed a significant contralateral difference in group-level handedness (Mean Handedness Index, MHI) between left-biased *PT versus* non-left-biased *PT* groups in newborn (Study 1: Fig. 2, *p* < .036) baboons for gestural communication only but not for non-communicative manipulation (*p* > .10). Additional one-tailed one-sample *t*-test highlighted a significant degree of positive MHI (i.e., right-handedness) in the left-biased *PT* group in newborn (Study 1: Fig. 2, *p* < .005) baboons for gestural communication only, but not for non-communicative manipulation or the non-left-biased *PT* groups (*p* > .10). ** *p* < .005; * *p* < .036, *ns*: non-significant. Source data are provided as a Source Data file.

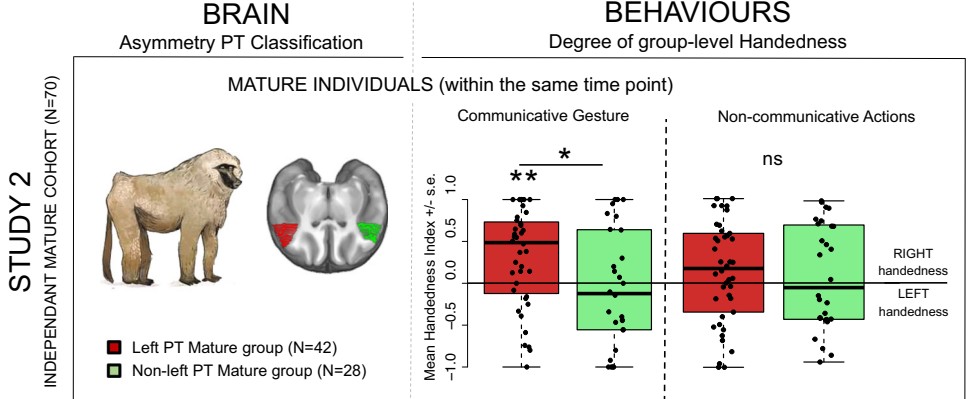

**Fig. 3 | Gestural communication handedness in mature baboons according to their planum temporale (*PT*) asymmetry.** Difference in the degree of group-level handedness (Mean Handedness Index, M.HI) in mature baboons for the communicative gesture and for non-communicative action, between left-biased *PT* (*N* = 42 in red) *versus* non left-biased *PT* groups (*N* = 28 in green) as classified within the same mature age class. Positive values indicate right-hand preference, negative values left-hand preference. One-tailed two-sample *t*-tests showed a significant contralateral difference in group-level handedness (M.HI) between left-biased *PT versus* non-left-biased *PT* groups mature baboons (Study 2: Fig. 3, *p* < .015) for gestural communication only but not for non-communicative manipulation (*p* > .10). Additional one-tailed one-sample *t*-test highlighted a significant degree of positive MHI (i.e., right-handedness) in the left-biased *PT* group in mature baboons (Study 2: Fig. 3, *p* < .0015) for gestural communication only, but not for non-communicative manipulation or the non-left-biased *PT* groups (*p* > .10). ** *p* < .0015; * *p* < .015, *ns*: non-significant. Source data are provided as a Source Data file.

direction of handedness emergence for communicative gesture (but not for non-communicative action). This longitudinal study suggests thus the first directional relationship across lateralisation development in primates from early brain structure to its related communicative function. In fact, we cannot suspect the opposite "chicken-egg" hypothesis given the lateralisation of communicative gesture did not precede the emergence of *PT* asymmetry across development. Indeed, at time point 1, while *PT* asymmetries are already set up soon after birth, newborn baboons are unable to express any of the manual behaviours of interest. We had to wait for those behaviours and their lateralisation to emerge much later across development, namely at time point 2 (from 7 months old at the earliest) when the juvenile baboons are mature enough to express this manual repertoire and to interact socially with other baboons outside their mother.

It remains unclear whether this finding is explained by a causal relationship between early *PT* asymmetry in newborns, potentially acting as a neurobiological determinant, and the further development of handedness for gestural communication (but not for manipulation) once juveniles. Indeed early brain prediction of future behaviour across time does not necessarily mean causality. Alternatively, it is possible that one or several external factors may affect the direction of both *PT* and gestural asymmetry independently, such as genetic- or environmental prenatal factors (e.g., foetal position), suggesting that there is no direct brain/behavioural link. Further investigation would be needed to investigate such potential external factors.

Further investigations would also explore the ontogeny of the frontal region to test whether an early asymmetry of Broca's homologue also exists in newborns, such as the planum temporale asymmetry described here, and whether it predicts the later

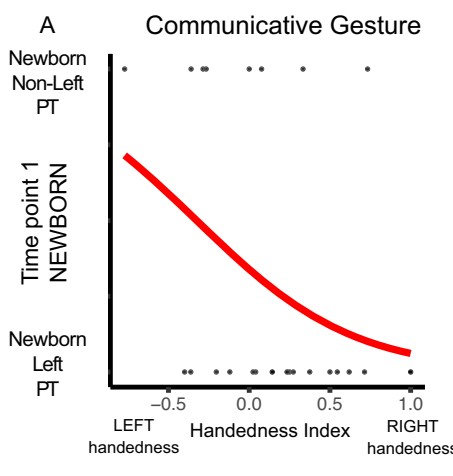
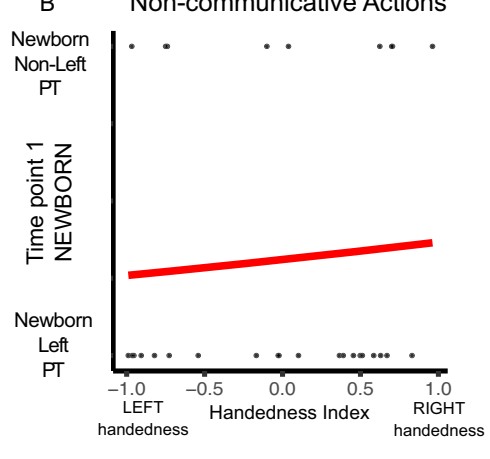

**Fig. 4 | Logistic regression curves: Handedness Index (HI) as function of early planum temporale (PT) asymmetry.** Positive values of HI indicate right-hand preference, negative values left-hand preference. Significant two-sided logistic regression model ($p < .044$) of the outcome probability of becoming left- or right-handed in (**A**) communicative gesture and in (**B**) non-communicative actions by the *PT* lateralisation classification at birth. Source data are provided as a Source Data file.

development of communicative gestures' lateralisation. Nevertheless, with previous and present findings from both frontal and temporal areas in nonhuman primates, it might be hypothesised that the early lateralisation for the planum temporale might be extended to a whole network of brain structures including thus Broca's area homologue. According to this latter hypothesis, such an early language-homologue network, ultimately related to the development of gestural communication specialisation in nonhuman primates, may also include its connected white matter fibre tracts, such as the arcuate fasciculus[18].

If the *PT* is involved in gestural communication in nonhuman primates, this region may not have solely evolved as a computational hub for complex sounds[19] but also seems to be specialised for communicative production. In fact, an alternative view depicts the *PT* as composed of functional subfields, including one performing motor processes[2]. Indeed, recent results in implanted human patients have shown that stimulation of the *PT* selectively disrupts speech production[20].

Given the lack of research in humans, further studies would investigate whether hand preference for communicative gestures specifically might be a better candidate than typical handedness measures related to non-communicative manual activities for predicting language lateralisation and/or structural *PT* brain asymmetries.

These collective findings suggest an evolutionary continuity between human and nonhuman primates in both the phylogeny and the ontogeny of their brain specialisation for communication. We therefore propose the gestural communication system, its intentional and referential cognitive properties, and its early structural brain specialisation as a shared conserved mechanism across primate ontogeny at the heart of human language-brain evolution, inherited from their common Catarrhini ancestor 25-36 million years ago.

## Methods

### Subjects

Study 1: Of our MRI cohort of 35 newborn baboons at time point 1 in which the *PT* structural asymmetry was measured in the study by Becker et al.[10], 30 could be observed longitudinally into their juvenile age class (i.e., time point 2). As only 3 subjects did not perform the data point of behaviours of interest in juvenile age, both behavioural and brain data were available for 27 subjects of this longitudinal cohort (10 females, 17 males; 4 days to 2 months of age prior to weaning and the full

maturation of myelin, synapses and cell bodies + one outlier at 165 days of age).

Study 2: Of the 93 mature baboons in which *PT* structural asymmetry was measured in the study by Marie et al.[8], both brain and behavioural data were available for 70 subjects (42 females, 28 males, age range = 2–26 years, mean age: $M = 11.8$ years, $SE = 0.76$).

### MRI cerebral acquisition

Study 1: In the newborn baboons' cohort, structural T1-weighted images were acquired on a 3 T Magnetom Prisma scanner (Siemens) using two 11 cm receive-only loop coils (see[9] for the full imaging protocol) thanks to a MPRAGE sequence (time repetition: 2500 ms; time echo: 3.01 ms; flip angle: 8°; inversion time: 800 ms; field of view: $103 \times 103 \times 102.4$ mm; isotropic voxel size: 0.4 mm³).

Study 2: In the mature baboons' cohort, structural T1-weighted images were acquired with a 3 T MEDSPEC 30/80 ADVANCE scanner (Bruker) using a Rapid-Biomed surface antenna (see[8] for the full imaging protocol) thanks to a MPRAGE sequence adapted to a female/young male (time repetition: 9.4 ms; time echo: 4.3 ms; flip angle: 30°; inversion time: 800 ms; field of view: $108 \times 108 \times 108$ mm; isotropic voxel size: 0.6 mm3) or to a larger adult male (same parameters except for field of view: $126 \times 126 \times 126$ mm and isotropic voxel size: 0.7 mm³).

### Brain measurements

*PT* structure measurements are related to previous published MRI data in baboons. In the newborn cohort (Study 1), measures of *PT* grey matter volume interhemispheric asymmetry originated from Becker et al.[10]. In the mature cohort (Study 2), measures of *PT* surface interhemispheric asymmetry originated from Marie et al.[8].

The *PT* surface variable provides a sufficient marker to infer interindividual *PT* variability in mature baboons[8] and its behavioural correlates (Study 2) regarding their bigger brain size and the larger sample size (N = 70) compared to newborn baboons. Documenting interindividual *PT* variabilities in smaller newborn brains with a much lower sample size (N = 27) is more sensitive when quantifying the entire *PT* (see[10]), thus including the underlying grey matter volume under the surface, especially when evaluating behavioural correlations as in the present study (see also[7] in chimpanzees).

Manual delineation of the *PT* utilised 'ANALYSE 12.0 (AnalyzeDirect)' software with a WACOM Cintiq 13HD graphics tablet, following established delineation instructions from previous MRI non-human

primate studies e.g.,[7–10,15,21]. In each baboon and in each hemisphere, tracing was conducted on coronal planes. The posterior edge of the *PT* was defined as the most caudal section displaying the Sylvian fissure. The anterior edge was delineated as the most anterior slice with a fully closed insula, coinciding with the Sylvian fissure's anterior disappearance due to the absence of a landmark defining the Heschl's gyrus. For each slice, tracing followed the Sylvian fissure's ventral edge from the medial to the lateral point. Specifically for grey matter results from the newborn baboons[10], graders traced to the grey-white matter boundary's most inferior point. When unclear, the imaginary extension of the Sylvian fissure helped distinguish the grey matter of interest from the dorsal gyrus. This process continued in the coronal plane until the Sylvian fissure vanished.

An asymmetry quotient (AQ) was computed of the left (L) and the right (R) *PT* structure: $AQ = (R-L) / [(R + L) \times 0.5]$, with the sign indicating the direction of asymmetry (negative: left hemisphere, positive: right hemisphere) and the value, its strength. Subjects were classified into two groups depending on the direction of *PT* asymmetry: left *PT* (AQ ≤ − .025) *versus* non-left *PT* (AQ > − .025)[10].

### Behavioural measurements

Behavioural measurements included both handedness for communicative gestures *versus* handedness for non-communicative bimanual object manipulation. Handedness for manipulative actions was assessed using the bimanual coordinated "Tube task"[22]. Regarding handedness for communicative gestures in baboons, the study focussed specifically on the 'hand slap' gesture, a behaviour previously identified as optimal for assessing the lateralisation of gestural communication in this species[13,14]. Indeed, the hand slap, employed for threatening or intimidating recipients, is not only the most common visual gesture in the baboon repertoire[11]; but is also clearly lateralised and unimanually performed in a goal-directed manner, particularly in agonistic contexts and similar postures of the sender[13,14]. A baboon's hand usage was recorded when it rapidly and repeatedly slapped or rubbed its hand on the ground, directed towards a conspecific or a human observer beyond its immediate reach. Notably, the recorded events were derived from multiple events ( > 5 events minimum per subject) documented across multiple sessions spanning from September 2020 to Mai 2021.

For each subject and for each of the behaviours, a handedness index was computed from the number of left-hand responses (L) and of the right hand responses (R): $HI = (R-L) / (R + L)$ with the sign indicating the direction of asymmetry (negative: left side, positive: right side) and the value, its strength.

### Statistics & Reproducibility

*PT* structural asymmetries data for newborn and mature subjects were derived from previously published MRI data[8,10]. Handedness data for mature subjects were also derived from previously published data[14] while handedness data for newborn subjects were newly collected for the present study. All subjects for whom both behavioural and brain data were available were included in this study. No statistical method was used to predetermine the sample size. Investigators were blinded to allocation during experiments and outcome assessment. As the contralateral direction of the effect (e.g., left brain -> right hand) was hypothesis-driven, one-tailed tests were used rather than two-tailed tests. The statistical testing was conducted by the following steps:

1. Normal distribution was tested with a Shapiro–Wilk test.
2. One-tailed two-sample *t*-tests tested differences in group-level handedness (Mean Handedness Index, MHI) between left-biased *PT versus* non-left-biased *PT* groups in both newborn (Study 1) and mature baboons (Study 2) for gestural communication and for non-communicative manipulation.
3. One-tailed one-sample *t*-test tested the degree of MHI in the left-biased *PT* group and in the non-left-biased *PT* groups in both

newborn (Study 1) and mature baboons (Study 2) for gestural communication and for non-communicative manipulation.
4. For Study 1, a logistic regression was performed to investigate the effects of the *PT* lateralisation classification (i.e. left-biased *PT versus* non-left-biased *PT* groups) at birth on the likelihood of becoming left or right-handed in communicative gesture when juvenile, controlling for non-communicative handedness (Fig. 4A). In other words, if the predictor variable "*PT* lateralisation classification" contributes significantly to the model.
5. For Study 2, a logistic regression was performed to investigate the effects of the "*PT* lateralisation classification" in mature subjects on the likelihood of being left or right-handed in communicative gesture, controlling for non-communicative handedness. In other words, if the predictor variable "*PT* lateralisation classification" contributes significantly to the model. Step 4. and 5. were also conducted to test in the same manner the effect of "*PT* lateralisation classification" on the likelihood of being left or right-handed for non-communicative actions.

### Ethical statement

All animal procedures were approved by the "C2EA−71 Ethical Committee of neurosciences" (INT Marseille) under the number APAFIS#13553-201802151547729 v4, and have been conducted at the Station de Primatologie under the number agreement C130877 for conducting experiments on vertebrate animals (Rousset-Sur-Arc, France). All methods were performed in accordance with the relevant French law, CNRS guidelines and the European Union regulations (Directive 2010/63/EU).

### Reporting summary

Further information on research design is available in the Nature Portfolio Reporting Summary linked to this article.

## Data availability

Source data are provided with this paper as a Source Data file at https://doi.org/10.6084/m9.figshare.24794076.

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

## Acknowledgements

We would like to thank Lola Rivoal for the baboon illustrations and the interns who helped us collecting the behavioural data: Amélie Picchiottino, Solène Brunschvig, Célina Jannas, Lou Cadau and Auban Galvin. We are grateful to Pierre Cherel for statistical power analysis and to Nico Scherf for statistical advice; to the LPC & ILCB managers: Frederic Lombardo, Colette Pourpe, Johannes Ziegler, Philippe Blache, Nadera Bureau; to the staff of the INT, Emilie Rapha, and of the Station de Primatologie CNRS for the care of the baboons: Lucie Faccin, Romain Lacoste, Pascaline Boitelle, Janneke Verschoor, Pau Molina, Jean-Christophe Marin, Valérie Moulin, Richard Francioly, Célia Sarradin, Brigitte Rimbaud, Magali Ghirart, Sebastien Guiol, Gregory Desor, Slaveia Garbit, Annie Massa and Christophe Arnoult, as well as the animal care staff from Animalliance. The project has received funding from the European Research Council under the European Union's Horizon 2020 research and innovation programme grant agreement No 716931—GESTIMAGE—ERC-2016-STG (A.M.), as well as French National Research Agency ANR-23-CE28-0029-01 (BABONTO) (A.M.), ANR-16-CONV-0002 (ILCB) (A.M., Y.B.), ANR-17-EURE-0029 (NeuroSchool) (A.M., Y.B.), the Excellence Initiative of Aix-Marseille University via A*Midex funding (AMX-19-IET-004) (A.M., Y.B.) and Fondation Fyssen (Y.B.). This work was performed in the Centre IRM-INT (UMR 7289, AMU-CNRS), platform member of France Life Imaging network (grant ANR-11-INBS-0006).

## Author contributions

Designed research: A.M. Performed research: Y.B., S.B., J.S., L.V., O.C., L.R., A.C., J.L.A., B.N., and A.M. Analysed data: Y.B., R.P., D.M., R.M., and A.M. Wrote the paper: Y.B. and A.M.

## Competing interests

The authors declare no competing interests.
