## [Peer Review File · Nature Communications]

Planum Temporale asymmetry in newborn monkeys predicts the future development of gestural communication's handednessREVIEWER COMMENTS

Reviewer #1 (Remarks to the Author):

This is a very interesting study, adding to the literature bearing on human language evolution, in particular on the role hemispheric asymmetries may play in communication. Building on their previous work, the authors present evidence for an early asymmetry (located around the Planum Temporale), associated with communicative gestures, in baboons. The study touches on long-standing issues in the field of language evolution, such as gestural origins, hemispheric asymmetries, and the importance of (neural) markers emerging early in ontogeny and predicting adult outcomes.

I am in favor of publication, but would like the authors to contextualize their current findings better by discussing how they relate to their previous work (e.g., their recent eLife study). I would also ask them to discuss the issue of causation vs correlation in the relation between hemispheric asymmetry and communicative behaviors.

Reviewer #2 (Remarks to the Author):

This study reports the structural asymmetry of the PT in newborn baboons, with such asymmetry found to be associated with communicative gestures but not with non-communicative actions. The authors propose that this PT asymmetry is a pre-wired neural signature of communicative function in monkeys that is shared by human language, which challenges the notion that the PT asymmetry is a human-specific marker for language.

Main points:

- 1) It is unclear how the PT was spatially defined in newborn baboons. The current findings were discussed with reference to previous studies on PT asymmetry in monkeys and humans. It is important to determine whether the structural location of PT in this study can be compared to that of previous studies.
- 2) The authors examined PT asymmetry in both mature and newborn subjects, but it is unclear why the regression analysis between structural asymmetry of PT and communicative performance was conducted only in newborns. I wonder whether the specific relationship observed in newborns is sustained in mature subjects. Furthermore, the regression model did not include any controlling factors. If we expect to observe specific correlations between PT asymmetry and communicative gestures, several covariates should be taken into account. For example, such correlations should survive the partialling of non-communicative performance, and no relationship was observed for the opposite analysis.
- 3) In the discussion, the authors argue that this is a longitudinal study and draw a conclusion about the causal relationship between PT asymmetry and gesture communication. However, this study only examined the correlation between structural asymmetry of PT and communicative gesture at one time point. To establish a causal relationship, multiple measures of behavioral performance and brain structure features at different time points are required within a longitudinal study.

Minor points:

- 1) The sample size for the left PT and non-left PT newborns is very small, and the standard deviation is larger than the mean value. Using a t-test to examine the differences between the groups may not be appropriate.
- 2) It is unclear why the surface property of the PT was measured in mature subjects while gray matter volume was measured in newborns. Further clarification on this choice would be helpful.
- 3) The criterion of $AQ = -0.025$ was used for the classification of left-biased and right-biased PT groups, but this is not a typical criterion for defining brain asymmetry. An explanation for this choice would be helpful.

Reviewer #3 (Remarks to the Author):

The authors show that hemispheric lateralisation in baboons (in both neonates and adults) is associated with right handedness in gestural communication production. The authors suggest this shows a shared neural underpinning of communicative systems across primates, likely co-opted during human language evolution.

The paper is interesting and the data look convincing, but I have some queries about the interpretation.

First, the authors show that there is a relationship between neural asymmetry and handedness in communicative gestures but not other manual actions in the baboons. Why do humans then demonstrate handedness across motor tasks, not just communication/language relevant movements? In fact, do humans show strong handedness in gesture, more so than manual tasks like throwing etc? What, if anything, is restricted to communicative domains? These are interesting questions that in my view should be addressed as they impact the relevance of the finding.

Second, do the more lateralised baboon individuals have better communication? What is the advantage to being more lateralised if not? Can the authors consider how/why this lateralisation was present in the ancestral brain before language arose? Related, I think the intro needs some brief discussion of why humans exhibit this laterality and why has that developed through evolution. Whether monkey gestures have the same intentional properties has also been the subject of intense discussion over the years. In my view this controversy is worthy of some comment in the intro.

Methodologically, the communicative gesture data seem to be based only on one gesture (hand slap). It would help to have more information about this, what it is used for, how data was collected, how many were recorded per individual etc. I appreciate the authors are aiming for a brief report, but the reader needs to be able to evaluate the behavioural data in order to fully appreciate the conclusions.

Minor points

Lines 34-36. Clunky sentence, can you rephrase?

REVIEWER COMMENTS

Reviewer #1 (Remarks to the Author):

This is a very interesting study, adding to the literature bearing on human language evolution, in particular on the role hemispheric asymmetries may play in communication. Building on their previous work, the authors present evidence for an early asymmetry (located around the Planum Temporale), associated with communicative gestures, in baboons. The study touches on long-standing issues in the field of language evolution, such as gestural origins, hemispheric asymmetries, and the importance of (neural) markers emerging early in ontogeny and predicting adult outcomes.

Thank you very much for your well-disposed and constructive feedback which improves our manuscript.

Remark 1: I am in favor of publication, but would like the authors to contextualize their current findings better by discussing how they relate to their previous work (e.g., their recent eLife study).

Response 1: Thank you for this demand. We have now added two paragraphs to the discussion section:

“This finding is consistent with previous research in baboons and chimpanzees showing a correlation between handedness measures for gestural communication and lateralized “language-homolog” brain structure, including other regions of interest such as Broca’s area (8,9) (in baboons: Becker et al., 2022; in chimpanzees: Tagliabata et al., 2006; Hopkins & Nir, 2010; Meguerditchian et al., 2012). For instance, baboons preferring to communicate with their right hand have a deeper left-than-right Inferior Arcuate sulcus in its most ventral section (a sulcal border of Broca’s area homolog) than those preferring to communicate with their left hand and vice versa (9). However, because those studies were conducted within a single time point/age class, namely when both manual behaviors and brain structures were already fully lateralized and mature, they were unable to determine whether these brain asymmetries precede or follow the development of lateralized manual behaviors.”

and further below in the discussion section:

...

“Further investigation would also explore the ontogeny of frontal region to test whether (1) early asymmetry in Broca’s homolog also exists in newborn, just as the planum temporale, and (2) is predictive of later development of communicative gestures’ lateralisation. Nevertheless, with previous and present findings from both frontal and temporal areas in nonhuman primates, it might be hypothesized that the early lateralisation for the Planum Temporale might be extended to a whole network of brain structures including thus Broca’s area homolog. According to this latter hypothesis, such an early language-homolog network, ultimately related to the development of gestural communication specialization in nonhuman primates, may also include its connected white matter fibre tracts, such as the Arcuate Fasciculus (Becker et al., 2022).”

Remark 2: I would also ask them to discuss the issue of causation vs correlation in the relation between hemispheric asymmetry and communicative behaviors.

Response 2: Thank you for this important point, which requires more discussion. We added a full section in the discussion about (1) the issue of causation vs correlation and about (2) alternative hypotheses from our results.

..."Second, our present study in the other cohort of 27 developing baboons at two longitudinal time points from birth allowed us to address this question for the first time by testing this "egg-chicken" hypothesis. We found that early detection of structural *PT* asymmetry at the first time point, namely soon after birth, predicts, for time point 2, the direction of handedness emergence for communicative gesture (but not for non-communicative action). This longitudinal study suggests thus the first time directional relationship across lateralisation development in primates from early brain structure to its related communicative function. In fact, we cannot suspect the opposite "chicken-egg" hypothesis given the lateralisation of communicative gesture did not precede the emergence of *PT* asymmetry across development. Indeed, at time point 1, while *PT* asymmetries are already set up soon after birth, newborn baboons are unable to express any of the manual behaviours of interest. We had to wait for those behaviours and their lateralisation to emerge much later across development, namely at time point 2 (from 7 months old at the earliest) when the juvenile baboons are mature enough to express this manual repertoire and to interact socially with other baboons outside its mother.

It remains unclear whether this finding is explained by a causal relationship between early *PT* asymmetry in newborns, potentially acting as a neurobiological determinant, and the further development of handedness for gestural communication (but not for manipulation) once juveniles. Indeed early brain prediction of future behaviour across time does not mean necessary causality. Alternatively, it is possible that one or several external factors may affect the direction of both *PT* and gestural asymmetry independently, such as genetic- or environmental prenatal factors (e.g., fetal position), suggesting that there is no direct brain/behavioural link. Further investigation would be needed to investigate such potential external factors."

Reviewer #2 (Remarks to the Author):

This study reports the structural asymmetry of the *PT* in newborn baboons, with such asymmetry found to be associated with communicative gestures but not with non-communicative actions. The authors propose that this *PT* asymmetry is a pre-wired neural signature of communicative function in monkeys that is shared by human language, which challenges the notion that the *PT* asymmetry is a human-specific marker for language.

Thank you very much for your well-disposed and constructive feedback which improves our manuscript.

Main points:

Remark 1: It is unclear how the *PT* was spatially defined in newborn baboons. The current findings were discussed with reference to previous studies on *PT* asymmetry in monkeys and humans. It is important to determine whether the structural location of *PT* in this study can be compared to that of previous studies.

Response 1: Thank you for asking for clarification here. In fact, we actually used the *PT* data and results in both newborn monkeys (study 1) and mature monkeys (study 2) from our previous papers in which methods of delineation were fully described, namely "Becker et al., 2022 Brain Structure and Function" for newborn monkeys and "Marie et al., 2018 Cerebral Cortex" for mature baboons.

It would be indeed clearer and useful to remind this delineation method in the revision. We added then the following paragraph in the method section:

"Manual delineation of the PT utilized 'ANALYZE 11.0 (AnalyzeDirect)' software with a WACOM cintiq 13HD graphics tablet, following established delineation instructions from previous MRI non-human primate studies (e.g., Hopkins and Nir, 2010; Lyn et al., 2011; Meguerditchian et al., 2012; Marie et al., 2018; Becker et al., 2021; Becker et al., 2022). In each baboon and in each hemisphere, tracing was conducted on coronal planes. The posterior edge of the PT was defined as the most caudal section displaying the Sylvian fissure. The anterior edge was delineated as the most anterior slice with a fully closed insula, coinciding with the Sylvian fissure's anterior disappearance due to the absence of a landmark defining the Heschl's gyrus. For each slice, tracing followed the Sylvian fissure's ventral edge from the medial to the lateral point. Specifically for grey matter results from the longitudinal study (Becker et al., 2022), graders traced to the grey-white matter boundary's most inferior point. When unclear, the imaginary extension of the Sylvian fissure helped distinguish the grey matter of interest from the dorsal gyrus. This process continued in the coronal plane until the Sylvian fissure vanished."

Remark 2a: The authors examined PT asymmetry in both mature and newborn subjects, but it is unclear why the regression analysis between structural asymmetry of PT and communicative performance was conducted only in newborns. I wonder whether the specific relationship observed in newborns is sustained in mature subjects.

Response 2a: Thank you for pointing this out. We are happy to extend this analysis to mature subjects if needed (study 2). Note that the initial aim of this analysis was to evaluate the directional link of early PT asymmetry at birth and the likelihood of becoming left or right-handed in gestural communication later in life (study 1), once the gestural repertoire has developed. *This would not be possible in mature subjects (study 2) where the brain and the behavioural data was assessed at one single time point when adults.*

We have written in the result section:

"Similarly, for study 2, a logistic regression was performed to investigate the effects of the "PT lateralisation classification" in mature subjects on the likelihood of being left or right-handed in communicative gesture, controlling for non-communicative handedness. The predictor variable "PT lateralisation classification" was found to contribute to the model. The unstandardized Beta weight for the predictor variable; $B = (-0.842)$, $SE = 0.3965$, $Wald = 4.515$, $p < .016$. For non-communicative actions in mature subjects, the predictor variable "PT lateralisation classification" was not found to contribute to the model.»

Remark 2b: Furthermore, the regression model did not include any controlling factors. If we expect to observe specific correlations between PT asymmetry and communicative gestures, several covariates should be taken into account. For example, such correlations should survive the partialling of non-communicative performance, and no relationship was observed for the opposite analysis.

Response 2b: It is a good idea to include non-communicative handedness as a further control variables/covariates. When partialling out non-communicative handedness the effect stays exactly the same. We have included in line 85: "... controlling for non-communicative handedness"

Remark 3: In the discussion, the authors argue that this is a longitudinal study and draw a conclusion about the causal relationship between PT asymmetry and gesture communication. However, this study only examined the correlation between structural asymmetry of PT and communicative gesture at one time point. To establish a causal relationship, multiple measures of behavioral performance and brain structure features at different time points are required within a longitudinal study.

Response 3: Thank you for pointing out that the design of our paper was not clear. To make it clearer, we have thus split the paper in two separate studies :

- Study 1: Longitudinal data at two separate time points (Time point 1: when newborn + Time point 2: when juvenile). Brain measures were taken when newborn at Time point 1 (from 5 days old) before the ontogeny of any manual behavior of interest (including communicative gesture). Then, manual behaviour lateralisation measures were taken at Time point 2, as soon as the manual behaviour of interest emerged in the ontogeny (namely from 7 months old at the earliest when the baboons is mature enough to express this manual repertoire and to interact socially with other baboons outside its mother). Note that our interpretation of directional links between early PT asymmetry and further development of manual lateralisation for communicative gestures is based only on these measures in developing baboons.

-Study 2: Supplementary Data set in an independent cohort of mature baboons. These present additional results showing brain/behavior correlates in a separated group of mature subjects did not address the question of prediction/directionality from brain to behavior, since both types of data were collected within one single time point/age class, when both brain and behavioral variables were already fully developed. Thus, this study 2 only provides strong support to the link between PT and gesture asymmetry in monkeys at this specific older age class. More importantly, whereas study 1 showed that development of gestural asymmetry could be predicted at birth from early PT brain measures, this study 2 shows that this gesture/PT link seems to last later at adulthood.

Here are the 4 main changes we propose to make our paper design more easily understandable:

(1) We have redesigned the Figure 1 accordingly and splitted it into 3 figures.

(2) We have written at the end of the introduction:

*"In the present in vivo MRI and behavioural study in 97 baboons (*Papio anubis*) living in social groups, we have investigated 1) the potential longitudinal link of early structural PT asymmetry in 27 newborns at time point 1 with their future gestural communication's lateralisation once juvenile at time point 2; 2) whether this PT/gesture asymmetry link exists in adulthood in a second cohort of 70 baboons."*

(3) We changed the title "*Planum Temporale asymmetry in newborn monkeys: an early brain marker of future gestural communication development?*" to reflect more cautious interpretation about potential causality.

(4) We also added a discussion about it and proposed alternative hypotheses from our results. See the updated discussion below:

"The results of our studies in both 70 mature baboons and in 27 newborn are straightforward and led to two main original findings:

First, in our cohort of 70 weaned mature baboons, we found that communicative gesture right-handedness, but not handedness for non-communicative manipulation, is associated with leftward structural PT asymmetry. This finding is consistent with previous research in baboons and chimpanzees

showing a correlation between handedness measures for gestural communication and lateralized "language-homolog" brain structure, including other regions of interest such as Broca's area (in baboons: 11; in chimpanzees: 10, 13, 14). For instance, baboons preferring to communicate with their right hand have a deeper left-than-right Inferior Arcuate sulcus in its most ventral section (a sulcal border of Broca's area homolog) than those preferring to communicate with their left hand and vice versa (11). However, because those studies were conducted within a single time point/age class, namely when both manual behaviours and brain structures were already fully lateralized and mature, they were unable to determine whether these brain asymmetries precede or follow the development of lateralized manual behaviours.

Second, our present study in the other cohort of 27 developing baboons at two longitudinal time points from birth allowed us to address this question for the first time by testing this "egg-chicken" hypothesis. We found that early detection of structural PT asymmetry at the first time point, namely soon after birth, predicts, for time point 2, the direction of handedness emergence for communicative gesture (but not for non-communicative action). This longitudinal study suggests thus the first time directional relationship across lateralisation development in primates from early brain structure to its related communicative function. In fact, we cannot suspect the opposite "chicken-egg" hypothesis given the lateralisation of communicative gesture did not precede the emergence of PT asymmetry across development. Indeed, at time point 1, while PT asymmetries are already set up soon after birth, newborn baboons are unable to express any of the manual behaviours of interest. We had to wait for those behaviours and their lateralisation to emerge much later across development, namely at time point 2 (from 7 months old at the earliest) when the juvenile baboons are mature enough to express this manual repertoire and to interact socially with other baboons outside its mother.

It remains unclear whether this finding is explained by a causal relationship between early PT asymmetry in newborns, potentially acting as a neurobiological determinant, and the further development of handedness for gestural communication (but not for manipulation) once juveniles. Indeed early brain prediction of future behaviour across time does not mean necessary causality. Alternatively, it is possible that one or several external factors may affect the direction of both PT and gestural asymmetry independently, such as genetic- or environmental prenatal factors (e.g., fetal position), suggesting that there is no direct brain/behavioural link. Further investigation would be needed to investigate such potential external factors.

Further investigation would also explore the ontogeny of frontal region to test whether (1) early asymmetry in Broca's homolog also exists in newborn, just as the Planum Temporale, and (2) is predictive of later development of communicative gestures' lateralisation. Nevertheless, with previous and present findings from both frontal and temporal areas in nonhuman primates, it might be hypothesized that the early lateralisation for the Planum Temporale might be extended to a whole network of brain structures including thus Broca's area homolog. According to this latter hypothesis, such an early language-homolog network, ultimately related to the development of gestural communication specialization in nonhuman primates, may also include its connected white matter fibre tracts, such as the Arcuate Fasciculus (15)."

Minor points:

1) The sample size for the left PT and non-left PT newborns is very small, and the standard deviation is larger than the mean value. Using a t-test to examine the differences between the groups may not be appropriate.

Response: In fact, we chose t-test since the data follow a normal distribution as assessed by a Shapiro-Wilk test. For communicative handedness: $W = 0.97332$, $p\text{-value} = 0.6719$; and for Asymmetry Quotient: $W = 0.98821$, $p\text{-value} = 0.9679$.

2) It is unclear why the surface property of the PT was measured in mature subjects while gray matter volume was measured in newborns. Further clarification on this choice would be helpful.

Response: Thank you for this question. The reason is quite simple: we just capitalized on previous published works reusing already available data & results, namely measures of PT surface (“the top of the PT iceberg”) in mature baboons (see Marie et al., 2018 Cerebral Cortex) and measures of PT grey matter volume (the full iceberg) in newborn baboons (see Becker et al., 2022 Brain Structure and Function).

We added thus a paragraph in the method section explaining the reasons for the difference of degree of PT quantification (from surface to full grey matter) between adult and newborn baboons.

"The PT surface variable provides a sufficient marker to infer inter-individual PT variability in adult baboons, regarding their bigger brain size and the larger sample size of Marie et al.'s study (12) compared to Becker et al.'s study in newborn baboons (8). In this latter study, documenting inter-individual PT variabilities in smaller newborn brains with a much lower sample size (N=27) necessitates a quantification of the entire PT, including thus the underlying grey matter volume under the surface, especially for evaluating behavioural correlations as in the present study (see also Hopkins&Nir, 2010 in chimpanzees)."

3) The criterion of $AQ = -0.025$ was used for the classification of left-biased and right-biased PT groups, but this is not a typical criterion for defining brain asymmetry. An explanation for this choice [YB2] would be helpful.

Response: Thank you for this inquiry. In fact, this AQ threshold is used as a standard convention in our comparative field on brain asymmetry, initially from humans and apes studies (see for example Hopkins and Nir (2010)), we have extended to our monkey works. We have thus added this reference Hopkins and Nir (2010) for the AQ threshold.

Reviewer #3 (Remarks to the Author):

The authors show that hemispheric lateralisation in baboons (in both neonates and adults) is associated with right handedness in gestural communication production. The authors suggest this shows a shared neural underpinning of communicative systems across primates, likely co-opted during human language evolution.

Thank you very much for your well-disposed and constructive feedback which improves our manuscript.

The paper is interesting and the data look convincing, but I have some queries about the interpretation.

COMMENT 1:

First, the authors show that there is a relationship between neural asymmetry and handedness in communicative gestures but not other manual actions in the baboons. Why do humans then demonstrate handedness across motor tasks, not just communication/language relevant movements? In fact, do humans show strong handedness in gesture, more so than manual tasks like throwing etc?

What, if anything, is restricted to communicative domains? These are interesting questions that in my view should be addressed as they impact the relevance of the finding.

Response: Thank you for bringing this up. This is a very interesting question.

In humans, there are indeed few studies investigating handedness for gestural communication specifically, in comparison to typical measures of non-communicative handedness. Those studies have found that gestural communication elicited also right-hand dominance, including (a) signing in deaf people, (b) “co-speech gestures” (i.e., manual movements produced simultaneously when talking), and (c) pointing gestures by infants during speech development (reviewed in Meguerditchian et al., 2011a). In infant specifically, degree of right-handedness for preverbal gestures has been shown to (1) rise first in the development and be more pronounced than handedness for manipulation (Blake et al., 1994; Bonvillian et al., 1997; see also Fagard, 2013; Cochet & Vauclair, 2010) and (2) increase when the lexical spurt occurs in children contrary to manipulation handedness (Cochet et al., 2011), suggesting, just like baboons, a potential dissociation of lateralisation between gestural communication and non-communicative measures of manual activities.

It must be noted that, given our results in baboons, we are currently investigating this question in human deafs, hearing participant and patients in our team, to test whether handedness for gestural communication may constitute a better predictor of language lateralisation, and of PT structural asymmetry than typical non-communicative handedness measures (see preliminary results in our conference talk: Meguerditchian, A., & Trebuchon, A. (2019, september). The gestural origins of brain specialization for language: How studies in baboons inspired research on epileptic patient. Escape, Tererife, Spain).

We have thus included this mention in our discussion section:

“Given the lack of research in humans, further studies would investigate whether hand preference specifically for communicative gestures might be a better candidate than typical handedness measures related to non-communicative manual activities for predicting language lateralisation and/or structural PT brain asymmetries.”

We also include in the introduction the question of comparative handedness across primates and the specific signatures of gestural communication’s lateralisation:

“A potential relevant functional candidate to test is gestural communication given its shared “domain-general” intentional/goal-directed and referential properties with language (3). Interestingly, just like in humans and chimpanzees (9), baboons’ communicative gestures elicited specific manual lateralisation pattern in comparison to non-communicative manual activities. Namely, handedness for gestures showed not only an increased use of the right-hand at the population-level but also different patterns of manual lateralisation at the individual level (for a review: 9). Such specific manual lateralisation for gestural communication – but not handedness for non-communicative actions - was found associated with brain asymmetries of homologous key language structures, such as Broca’s area, in chimpanzees (10) and in baboons (11).”

COMMENT 2:

Second, do the more lateralised baboon individuals have better communication? What is the advantage to being more lateralised if not? Can the authors consider how/why this lateralisation was present in the ancestral brain before language arose? Related, I think the intro needs some brief discussion of why humans exhibit this laterality and why has that developed through evolution.

Response 2: Ok. We have added in the introduction some aspects of the selective pressures and adaptive value of hemispheric lateralisation. We couldn't address all those interesting related questions given how modest our present contribution is with respect to those specific "adaptive value" questions or link between "gesture performance" and lateralisation, especially in a short note. We had to make the choice to rather focus on the theoretical and evolutionary implications for language of such brain/behavior correlates in baboons.

We have nonetheless included in the introduction:

"Its specific adaptative value for language is unclear although lateralisation may constitute a fitness advantage. When it comes to the brain, there is a significant optimization in the accessible neural resources, especially for a high demanding cognitive resources like language. One hemisphere can perform a task, while simultaneously the other hemisphere oversees another task (4 Rogers, 2021).

COMMENT 3:

Whether monkey gestures have the same intentional properties has also been the subject of intense discussion over the years. In my view this controversy is worthy of some comment in the intro.

Response 3: That's a good question and a considerable debate which would not be addressed by our present contribution. To refer to this specific but key debate, we have thus also included the more consensual term of "goal-directed" communication which corresponds to the low-level definition here of the first order of "intentional communication". Line 43:

"...shared "domain-general" intentional/goal-directed and referential properties with language"

Methodologically, the communicative gesture data seem to be based only on one gesture (hand slap). It would help to have more information about this, what it is used for, how data was collected, how many were recorded per individual etc. I appreciate the authors are aiming for a brief report, but the reader needs to be able to evaluate the behavioural data in order to fully appreciate the conclusions.

Response: Thank you very much for this point. We have now added a paragraph in the method section:

"Regarding handedness for communicative gestures, the study focussed specifically on the 'hand slap' gesture, a behaviour previously identified as optimal for assessing the lateralisation of gestural communication in this species (20,21). Indeed, the hand slap, employed for threatening or intimidating recipients, is not only the most common visual gesture in the baboon repertoire (22); but is also clearly lateralized and unimanually performed in a goal-directed manner, particularly in agonistic contexts and similar postures of the sender (9). A baboon's hand usage was recorded when it rapidly and repeatedly slapped or rubbed its hand on the ground, directed towards a conspecific or a human observer beyond its immediate reach. Notably, the recorded events were derived from multiple events (> 5 events minimum per subject) documented across multiple sessions spanning from September 2020 to Mai 2021."

Minor points

Lines 34-36. Clunky sentence, can you rephrase?

Response: Yes, we have rephrased “Nevertheless, its functional implication in monkeys and its potential shared properties to any language feature related to *PT* asymmetry in humans remain unknown”. to:

*“...indicating that this brain feature is not language- or human-specific. It may also suggest a shared signature of a common ancient cognitive process at the heart of language evolution (7). But the function of such a structural *PT* asymmetry in monkeys remains unclear.”*

REVIEWERS' COMMENTS

Reviewer #1 (Remarks to the Author):

I thank the authors for taking my comments on board. I am satisfied with their revisions, and think the current version of the manuscript is ready for publication.

Reviewer #2 (Remarks to the Author):

The authors have done well in addressing my concerns. I recommend this study for publication.

Reviewer #3 (Remarks to the Author):

In my first review my comments largely related to broader evolutionary/functional questions that I think are essential to consider to give the findings appropriate context. The authors have responded well to all these points and included additional background information/theoretical context where necessary, while keeping mindful of space limits and the scope of the current paper. I think the paper will make an important contribution to the literature.